# Thermal Comfort in the Built Environment: A Digital Workflow for the Comparison of Different Green Infrastructure Strategies

Stefano Cascone *🔾 and Alessia Leuzzo 🔾

Department of Architecture and Territory, Mediterranea University of Reggio Calabria, Via dell'Università n. 25, 89124 Reggio Calabria, Italy
* Correspondence: stefano.cascone@unirc.it

**Abstract:** The green transformation of the built environment is aimed at improving sustainability and can be supported by digitalization, which has become a significant tool to support the supply, integration, and management of information throughout the construction life cycle. In addition, climate change highly affects human comfort in the built environment and different strategies should be evaluated for adapting cities. This paper developed a digital workflow by integrating existing tools (i.e., Grasshopper, Ladybug, Honeybee, and Dragonfly) to evaluate how different green infrastructure strategies affected the thermal comfort by reducing the UTCI. The workflow was applied to a typical historical urban context (Catania, South of Italy), consisting of a square surrounded by three-floor buildings. Three basic scenarios were created that depended on the pavement material used in the built environment: a black stone pavement (reference material from Mount Etna), a permeable pavement, and grass. These three scenarios were combined with different green infrastructure strategies: tree pattern on the square, green walls and green roofs on the surrounding buildings, and the integrations of all these above-mentioned strategies. The results demonstrated that the integration of different green strategies (a grass square instead of pavement, with trees, and green walls and green roofs) increased the thermal comfort by reducing the UTCI by more than 8 °C compared to the existing urban context (black stone pavement and building envelope). However, this temperature reduction was highly affected by the location of the human body into the urban context and by the evaporation rates from vegetation. The workflow developed will be useful for designers to evaluate the effectiveness of different green strategies during the early-design stage in mitigating and adapting cities to climate change.

**Keywords:** early-stage design; parametric model; climate change adaptation; human wellbeing; greenspaces; UTCI





## 1. Introduction

The exposure of cities to climate change is having an effect on human health because more than half of the world population currently lives in urban areas, which is leading to a fast reduction in green urban land [1]. The urban heat island phenomenon can result in temperature variations up to 8 °C among cities and their nearby suburban and rural areas [2,3]. Recently, researchers and designers suggested the use of reflective surfaces, cooling materials, and vegetation for considerably increasing the environmental urban conditions, decreasing the radiant temperature, improving the natural ventilation, and mitigating the urban heat island effects [4,5].

The ability of urban spaces to supply outdoor thermal comfort is important for urban livability, wellbeing, social cohesiveness, and friendliness, and it is a crucial concern in high-density urban areas [6]. However, a complete description of thermal comfort cannot be provided by only assessing micro-meteorological variables (direct and indirect solar radiations, mean radiant temperature, humidity, and wind) because it is influenced by the urban design, such as morphology (urban canyon), materials properties (albedo) and

dissipation surfaces (green cover, water surfaces, and soil), which is the most impactful element on the microclimate [7,8].

In response to these challenges, urban design strategies can be developed to decrease urban temperature and increase wellbeing and resilience in towns [9,10], also integrating digital technologies and sustainability assessment schemes [11]. These strategies comprise of green infrastructures, such as city parks, pocket gardens, tree patterns, and green roofs and walls, that are considered significant factors in regulating the micro and local climate conditions of the built environment, enabling evaporative cooling processes on building surfaces and/or in open spaces [12,13]. Urban green areas suggest a variety of ecosystem benefits, including reducing noise, air, and water pollution, lowering air temperature, and accommodating recreational activities [14]. In addition, vegetation intercepts short and long wave solar radiation and enhances the microclimate due to evapotranspiration and shading effects that alter the heat budget of the surrounding air and surfaces [15]. Radiation interception is due to canopy parameters while evapotranspiration is due to water content flow through the soil–vegetation–air mechanism, leading to lower air temperatures and improved outdoor comfort levels [16,17].

The idea of green infrastructure has been proposed as a rational planning entity to improve urban green areas [18]. It can be deemed to include all natural, semi-natural, and non-natural networks of multifunctional environmental structures in, around, and among urban areas, at all spatial levels [19]. Green infrastructure underlines the value and the amount of urban and suburban green areas, their multifunctional character, and the significance of interconnections among environments [20]. If a green infrastructure is proactively designed, established, and preserved, it has the ability to drive urban growth by supplying a support for economic development and environment preservation [21]. Such a proposed methodology would suggest many chances for its combination with urban growth, environmental management, and community wellbeing [22]. Even though there is not a unique meaning for green infrastructure system, Dong et al. [23] considered hubs and links as its key elements. Hubs are the areas with a natural value that support environments for biodiversity. Corridors, which work as connections that link hubs, allow the passage of resources, data, and species between hubs, and their connectivity concerns the capability of species to move among areas.

Despite the growing research on urban microclimates, determining measurable connections between green infrastructures and human comfort is becoming a promising research field. Measurement techniques are important to examine the outdoor thermal comfort and numerous indicators have been established in recent years [24]. Amongst these, one of the most recently developed indexes for thermal comfort evaluation is the universal thermal climate index (UTCI), which is based on a one-dimensional modelling of human physiological answers to climatic conditions involving the thermal issue, and which is controlled by multidimensional variables [25]. This model is expected to be applicable for all climatic conditions, seasons, and genders. UTCI takes the input variables of wind speed, relative humidity, air temperature, radiant temperature (typically counting the solar radiation), and it uses these inputs in the human energy balance to estimate an equivalent temperature that shows the heat or cold stress felt by the human body. The calculation of the physiological response to the meteorological input is built on a multi-node model of human thermoregulation, which is enhanced with a clothing model [26].

After analyzing previous research on the existing parametric model developed using Grasshopper and its plug-ins to evaluate the effect of urban vegetation on outdoor thermal comfort, this study aims at developing a parametric workflow by integrating different existing Grasshopper plug-ins, i.e., Ladybug, Honeybee, and Dragonfly, to evaluate the UTCI. The three scenarios analyzed comprised different pavement types: a stone pavement (reference material), a permeable (cool material plus grass) pavement, and grass. Each of these scenarios was then coupled with different green infrastructure strategies, i.e., tree pattern, green roofs and green walls on the building envelope, and the integration of the all the above-mentioned green strategies. Catania was chosen as case-study for the workflow

validation due to the historic city center and the dark material used for pavements and buildings, which coming from volcanic rocks produced by Mount Etna's eruptions.

This research can help designers in evaluating the outdoor thermal comfort during the early-design stage by comparing the effect of different green infrastructure strategies.

## 2. Background

Environmental modelling allows for the assessment of the effects of urban vegetation on the microclimate and thermal comfort [27]. Several researchers created microclimate comfort charts using Grasshopper (which is an algorithm-centered plug-in for Rhinoceros, a 3D modeling software) as it supports the visual programming interface [28,29]. Linking Grasshopper with Rhinoceros could refine most of the sophisticated applications for architecture and urban design by creating complex parametric geometries and models [30].

Grasshopper allows designers with no scripting knowledge to control, manipulate, and envision information through its graphical user interface. Within this interface, the Outdoor Comfort Calculator is a component included with Ladybug (an environmental analysis plug-in for Grasshopper), where UTCI is calculated and visualized by means of the source code of the equations settled by The International Society of Biometeorology in an effort to measure human comfort [31]. Ladybug allows the importation of the data from the .epw file into the Rhinoceros modeler environment and the Grasshopper graphic algorithm editor. In addition, it proposes a range of significant information visualizations in graphic maps and 3D interactive pictures, which can help designers in producing more advised project evaluation at the early-design stages. In addition to Ladybug, the Honeybee plug-in links Grasshopper to various simulation tools to build energy and daylight models [32].

The literature review is focused on previous research developing a digital workflow in Grasshopper and its plug-ins to evaluate the thermal comfort in the built environment by adopting different green infrastructures. Some studies developed a digital workflow by analyzing a single green strategy and evaluating its effect on human thermal comfort. Lin et al. [33] focused on green facade influences on human thermal comfort optimization in a transitional area.

To assess the thermal comfort indexes, the authors implemented a sequence of processes utilizing Ladybug combined with Honeybee. While Lin et al. [33] only examined the impact of vertical green facades, Gholami et al. [34] improved the feasibility of designing pedestrian-level thermal comfort in a high-density historic city environment. Physiological equivalence temperature (PET) was assessed by employing a hybrid model developed in Python computer code. Three engines, i.e., EnergyPlus, Grasshopper, and OpenFOAM, were the model basis. Because Grasshopper cannot be coupled with EnergyPlus clearly, Honeybee was employed to link Open Studio, EnergyPlus, Radiance, and Daysim for an automatic method to reconstruct a weather file established on the current .epw file. Similarly, Fahmy et al. [35] observed the optimal soil usage factors to accomplish pedestrian thermal comfort. After finding the design factors' coefficients, Grasshopper was used for the optimization of the vegetation geometry based on these recommended coefficients. Aimed on modifying tree positions to enhance outdoor thermal comfort in road parking lots and related sidewalks, Milosevic et al. [36] suggested a method for establishing tree positions in road parking lots using Grasshopper and determining the UTCI using Ladybug. Finally, all UTCI numerical outcomes (in °C) were transferred from Grasshopper to Microsoft Excel. For each tree variation, Ladybug determined the UTCI values, and the amount of final numerical outcomes was identical to the amount of all tree dislocations.

Other studies analyzed the impact of green strategies at urban scales. For example, Hamdan et al. [37] examined the effect of urban design strategies on microclimates, such as vegetation shading. EnergyPlus and OpenStudio, combined with Grasshopper (Ladybug plug-in), were employed to create the UTCI comfort charts. The EnergyPlus in Grasshopper modelled the resulting meteorological conditions that are affected by the environmental variables, and the climatological information input from the .epw file was evaluated in the new framework. These climatic variables were affected by the vegeta-

tion shading. Lobaccaro et al. [38] examined the mutual relations between the various urban surfaces, such as vegetation components. To carry out the assessment of the local weather's characterization, the open-source weather assessment Ladybug plug-in was employed. The three-dimensional model of the case-study neighborhood was designed using Rhinoceros and Grasshopper. Grasshopper script was utilized to generate the trees. Finally, Perini et al. [39] found a technique for modeling climate management strategies and the consequences of vegetation on the microclimate and outdoor comfort. The UTCI calculation and visual information image tool were built in Grasshopper to merge the outputs from ENVI-met and TRNSYS.

Most of the previous studies used ENVI-met software to analyze the thermal comfort in the built environment with different green infrastructures. One of the first studies in this research field was conducted by Zölch et al. [40], who compared three green infrastructure alternatives, i.e., trees, green roofs, and green facades. The modeling methodology showed that the extent of cooling varied extensively among the examined scenarios. Trees worked better than green roofs and facades because of the larger shade their canopy offers. The significance of the heat mitigation did not appear as an explicit relationship with the amount of the green covering. One of the most pioneering efforts to investigate the significance of tree positions and arrangements was carried out by Zhao et al. [41], who used a microclimate numerical model to study how the layout of trees could help both individual households and residential communities. The flexibility of numerical simulations made it feasible to create and evaluate the outdoor microclimates and thermal comfort under a broad variety of tree positions and arrangements. Finally, Schibuola and Tambani [42] determined the physiological equivalent temperature (PET) with the ENVI-met software to assess outdoor thermal comfort. To this end, several mitigation approaches were evaluated for the entire summer period. Green and cool roofs offered the best performances. Green wall technology was the most efficient in all urban shapes analyzed, occasionally achieving the neutral comfort level.

This background showed that designing accurate models for urban environments is extremely difficult because green spaces significantly differ from buildings in their radiative, thermal, and drag properties. However, Grasshopper and Ladybug enabled the users to explore the relationship between environmental data (e.g., UTCI) and urban design through numerical and graphical results. Compared to other models used for outdoor thermal comfort calculation (e.g., RayMan and ENVI-met), the ease of 3D modeling and the higher resolution of visual representations of urban environments are noticed advantages of Grasshopper. In addition, none of this previous research analyzed the impact of different green strategies on thermal comfort at the early-design stage.

## 3. Methodology

### 3.1. Scenarios Definition

The methodology started from a generic workflow provided by the Ladybug (LB), Honeybee (HB), and Dragonfly (DF) Grasshopper plug-ins, which help to create new workflows by combining already existing and new self-programmed components. Specifically, in this research, different components of the authoring software were used (Figure 1). Figure 2 is a diagram that describes the methodology used in this study.

The experimental comparison focused on several factors contributing differently within the same urban context. For this purpose, fixed and variable conditions were considered. The fixed conditions were the following: (1) the specific urban weather in Catania (southern Italy, Mediterranean climate); (2) the urban morphology (the analyzed area is a square of 2000 m$^2$ surrounded by eight buildings of the same height (9 m, 3 floors), which is a typical urban context in historical cities); and (3) the location of human body in the analyzed area (the human body can be used as a thermal comfort sensor within the workflow).

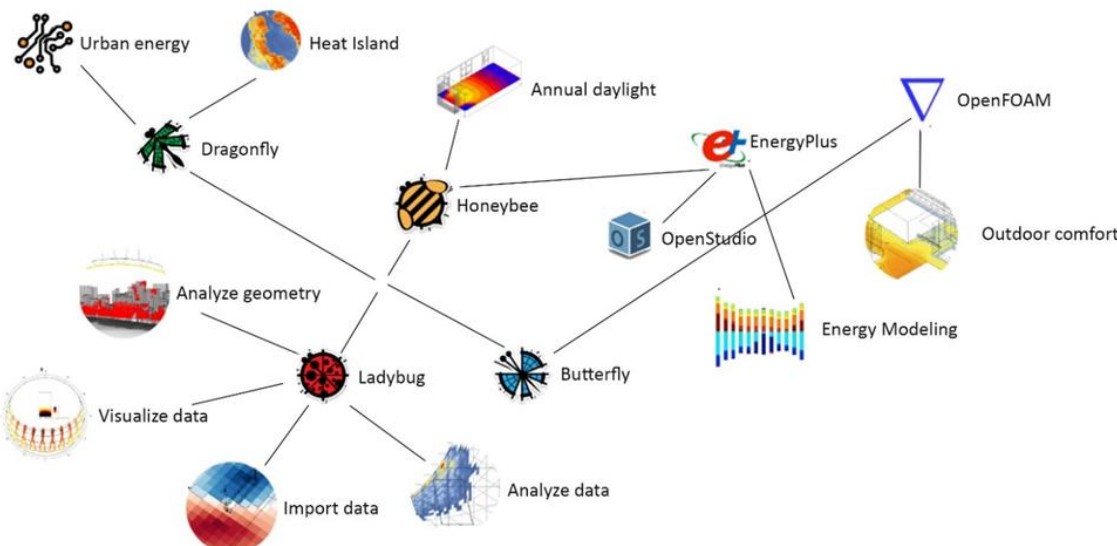

**Figure 1.** Map of the Ladybug tools and the fundamental components considered.

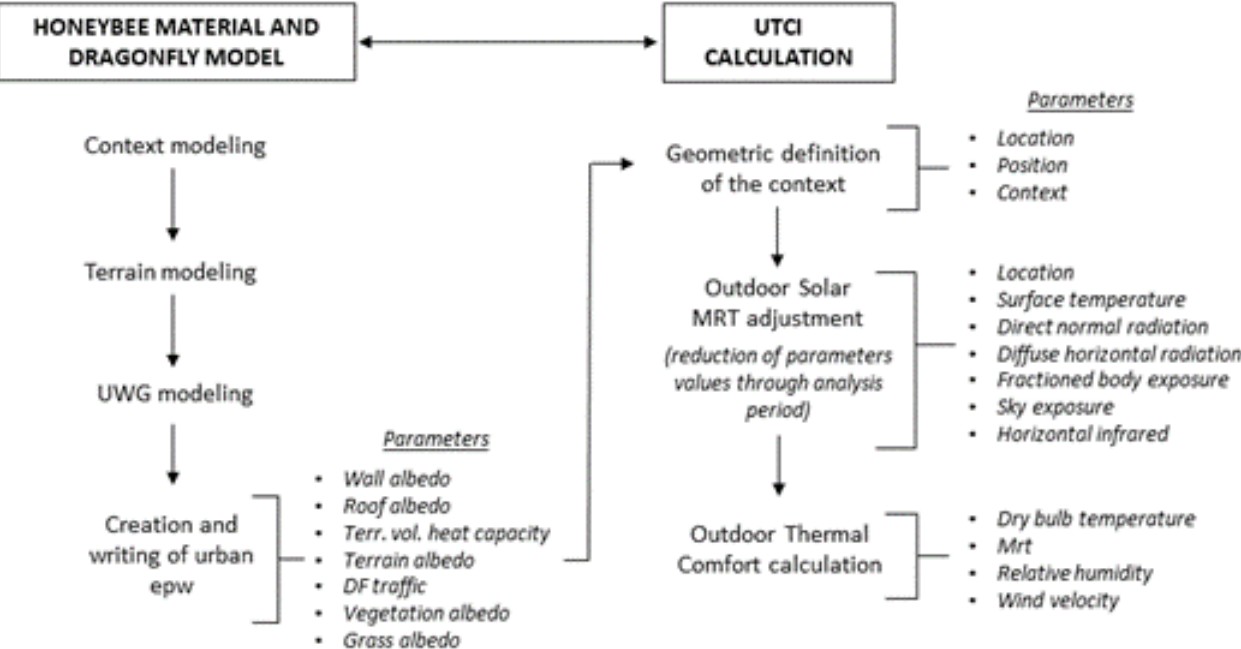

**Figure 2.** Methodology diagram.

The variable conditions in the workflow were the following: (a) the square pavement materials, which was either black stone pavement (Sp, which is the reference material), high-reflective permeable pavement (Pp), or grass; (b) the presence/absence of equally distributed trees on the square; (c) green roof and green wall systems (Gr/Gw) integrated into the envelope of the surrounding buildings, which are traditionally made from black stone (the reference material for the surrounding building envelope); and (d) the combination of all the above-mentioned strategies. The 12 scenarios generated in the workflow are showed in Figure 3.The black stone used for the pavements and the building envelope is the traditional material extracted from volcanic rocks produced by Mount Etna.

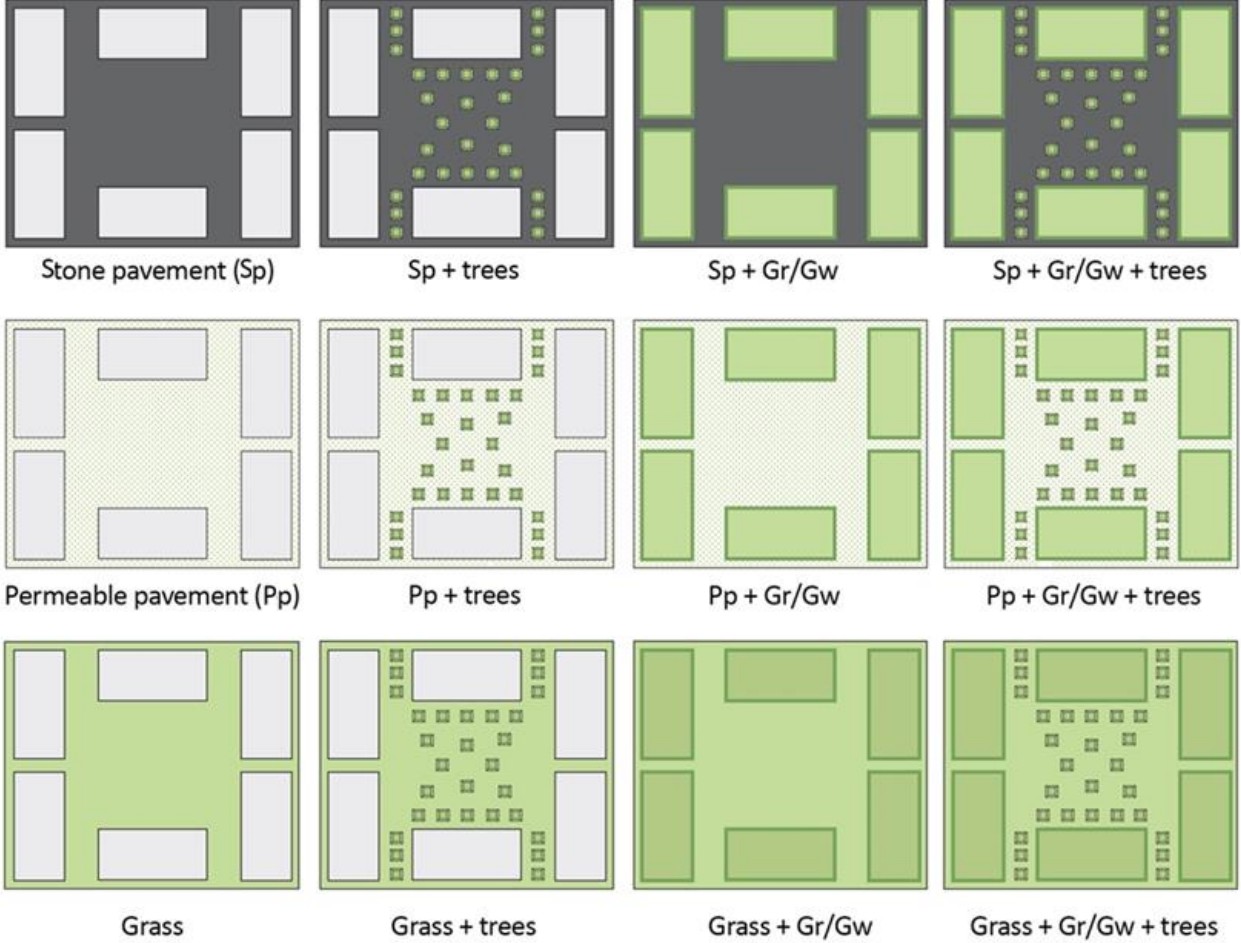

**Figure 3.** Schematic representation of the 12 scenarios generated in the workflow.

### 3.2. Workflow Phases

The workflow was divided in to two phases. The first phase was dedicated to the design of the scenarios and the definition of the geometrical and material characteristics, as they were important factors to determine outdoor comfort. The second phase focused on the calculation of the mean radiant temperature (MRT) and the universal thermal climate index (UTCI) on the human body placed into the urban context.

Specifically, the first workflow phase contained two sub-phases that used the Honeybee and Dragonfly plug-ins (Figure 4, top) as follows: (1) urban context and terrain modeling to define the reference scenario (i.e., building geometries and materials, pavement area, and albedo, to determine the heat absorbed/reflected); (2) Urban Weather Generator (UWG) modeling and running, as an outcome of the context built during the previous sub-phase, to create the new EnergyPlus Weather (.epw) file by adding the site-specific weather conditions from the UWG model to the ones available in the Catania .epw file.

The second workflow phase also contained two sub-phases which used the Ladybug plug-in (Figure 4, bottom) as follows: (1) importing the newly generated site-specific model .epw file; (2) creating the reference geometry for the thermal comfort calculation, outdoor solar temperature adjustment to determine MRT, outdoor comfort calculation, and data plotting.

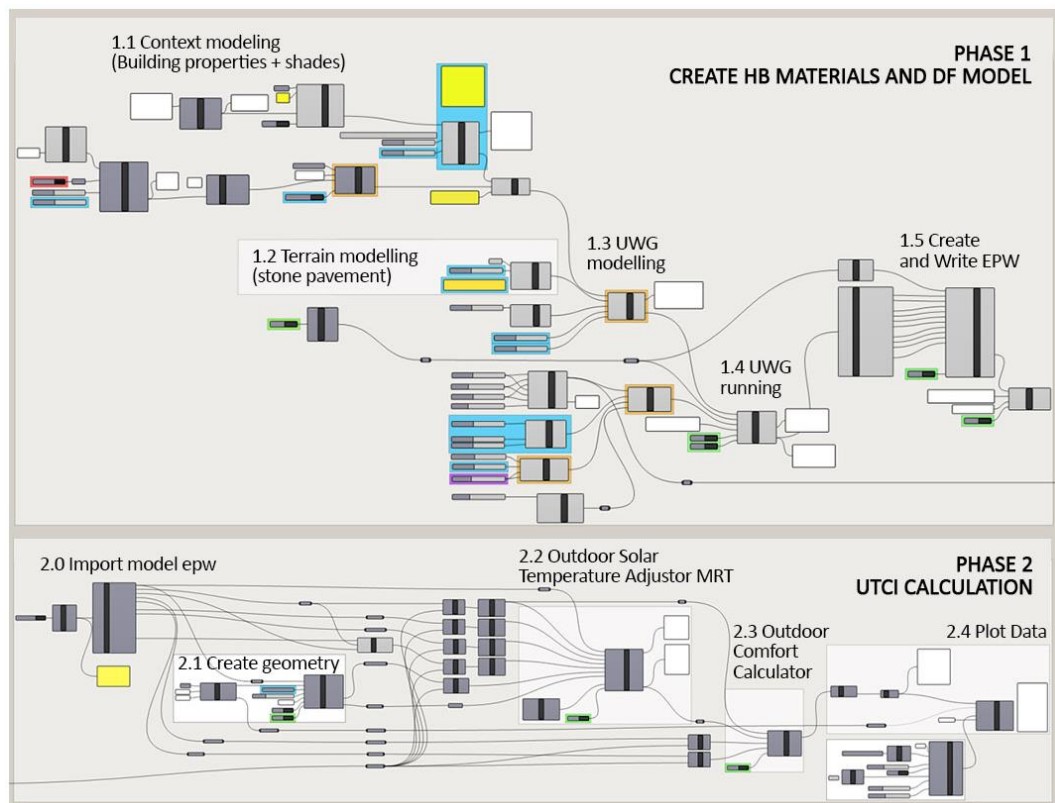

**Figure 4.** The two phases of the workflow: Phase 1 to create HB materials and DF model (**top**); Phase 2 for UTCI calculation (**bottom**).

### 3.3. UTCI Calculation and Physical Parameter Implementation

The UTCI calculation was based on a series of specific parameters primarily derived from the analysis period definition. Specifically, the parameters directly involved in the UTCI calculation are the air temperature, MRT (mean radiant temperature), relative humidity, and wind velocity. The air temperature is taken from the dry bulb temperature input within the Catania .epw file, an IGDG containing weather and energy data hour by hour, to which the analysis period was applied to include only useful data (day: 15 July, hour: 13:00). Relative humidity and wind velocity parameters were also built in this way, by using the "relative humidity" and "wind speed" parameters as inputs to the algorithm from the epw. The MRT considered: location, surface temperature, direct normal radiation, diffuse horizontal radiation, fractioned body exposure, sky exposure, and horizontal infrared. These parameters were found within the .epw (EnergyPlusWeather) file. The sky exposure was calculated by considering location, context, and position from the .epw file.

The horizontal infrared was calculated using three sub parameters as inputs in the Grasshopper component: sky cover, dew point temperature, and dry bulb temperature, which were in the .epw file.

In addition, the described parameters, even if they were taken from the .epw file and remained constant throughout the analysis period, were involved in creating the algorithm that defined each scenario. Although no specific data were input regarding wind velocity, the algorithm always considered the following influences on the thermal comfort of the human body: the geometric distribution of the surrounding buildings, their height, and the position of the human body (the distance of each building from the human body).

Finally, the UTCI calculation also was affected by the setting of the following initial parameters in the DF model generation: wall albedo (0.1 for buildings without green walls, 0.25 with green walls), roof albedo (no input because it considered the HB building vintages, 1980–2004 for buildings without green roof, 0.25 with green roof), volume heat capacity of the terrain, terrain surface (0.1 for black stone pavement scenario, 1.6 for semi permeable

pavement, 2.5 for grass), DF traffic parameters (8 W per area, similar to medium walking traffic and low car traffic), vegetation albedo, and grass albedo.

*3.4. Parameter and Scenario Definition*

The proposed scenarios included the square pavement type, the tree pattern, green roofs and walls on the surrounding buildings, and the integration of all these strategies. Three scenarios with different pavement materials were considered (i.e., stone, permeable, and grass) to which were added the green infrastructure strategies (i.e., trees, and green roofs and walls). The combination of these strategies created the different scenarios, as described below.

Furthermore, the analysis period was based on the hottest summer day and hour in Catania. Specifically, the choice of the 15 July was supported by a search into the weather conditions during the last 3 years, which resulted in this being the hottest day in Catania. With regards to the hour chosen, 13:00 was specifically chosen because it was easy to understand and control the specific thermal conditions and effects related to this time of the day.

Also, it is relevant to note that 13:00 is the hottest hour of the day, and thermal comfort is highly influenced not only by high temperature and humidity, but also by the very short shadows of this hour. As a consequence, this setting minimized the contribution of different shadows, allowing us to better understand whether the eventual reduction of temperature in the thermal comfort algorithm was due to the position of the human body directly in the shade of the tree foliage; at other times of the day, the longer shadows created by the surrounding buildings would have affected the results.

Each component used in the workflow was either classified as a fixed parameters describing fixed conditions (FpFc), fixed parameters describing variable conditions (FpVc), or variable parameters describing variable conditions (VpVc).

Four factors were considered as FpFc: (1) the Catania .epw file from the .epw map in the Ladybug archive; (2) the analysis period workflow component, starting on July the 15th at 1:00 p.m. (the hottest hour during a summer day in Catania), with one calculation timestep per hour; (3) the height (in meters) of the urban boundary layer during the daytime, which was set at 100 m to include the area close to the human body representing the height to which the urban meteorological conditions are stable and representative of the overall weather; (4) three floors per building, with three meters height for each floor, were considered as a "Solid" workflow component for the DF building in the context modeling.

Four factors were considered as FpVc: (1) DF reference .epw parameters representing the site properties and vegetation, as a fraction of the reference site (i.e., 0.0 for stone pavement, 1.0 for grass, 0.5 for permeable pavement); (2) in the scenarios containing trees, UWG can or cannot include vegetation; (3) in the DF Assign Building UWG properties component, the 1980–2004 (CBECS) vintage was assigned; (4) within scenarios with green roofs and walls, wall and roof albedo was 0.25, while in the reference scenario with stone material, wall albedo was 0.2 (automatically calculated from the HB Building Vintages of 1980–2004 assigned to the vintage characteristics).

VpVc were the workflow parameters and components depending on the combination of several green strategies into the three-pavement scenario (Figure 3), i.e., buildings, buildings and trees, buildings and Gr/Gw, buildings and Gr/Gw and trees.

In the scenarios, only the variables (FpVc and VpVc) were shown, while the FpFc were not reported, i.e., the initial Catania .epw file to run the DF Urban Weather Generator (UWG), the analysis period, the urban boundary layer during the daytime, and the number of floors of the surrounding buildings.

As a synthesis of the methodology built so far, Table 1 was created.

**Table 1.** Workflow components and parameters.

| | ContextShade | Terrain | ModelUWG | | UWGSimPar_(veg_par) | | | Ref.epwPar | HumanToSky |
|---|---|---|---|---|---|---|---|---|---|
| | uwg_is_veg | Albedo | Tree Cover | Grass Cover | Albedo | Tree Latent | Grass Latent | veg_cover | Context |
| **Scenario 1: Stone pavement** | | | | | | | | | |
| Buildings | False | 0.1 | 0 | 0 | 0 | 0 | 0 | 0 | Buildings + terrain |
| Buildings + trees | False (buildings) + True (trees) | 0.1 | None * | 0 | 0.16 | 0.7 | 0 | 0 | Buildings + terrain + trees |
| Buildings + Gr/Gw | False (buildings) + True (building facades) | 0.1 | 0 | 0 (terrain only) | 0.25 | 0 | 0.5 | 0.25 ** | Buildings (+ surfaces) + terrain |
| Buildings + Gr/Gw + trees | False (buildings) + True (trees and buildings facades) | 0.1 | None * | 0 (terrain only) | 0.16 | 0.7 | 0.5 | 0.25 ** | Buildings (+ building surfaces) + terrain + trees |
| **Scenario 2: Permeable pavement** | | | | | | | | | |
| Buildings | False | 0.25 | 0 | 0.5 *** | 0.25 | 0 | 0.5 | 0.5 | Buildings + terrain |
| Buildings + trees | False (buildings) + True (Trees) | 0.25 | None * | 0.5 *** | 0.20 | 0.7 | 0.5 | 0.5 | Buildings + terrain + trees |
| Buildings + Gr/Gw | False (buildings) +True (buildings facades) | 0.25 | 0 | 0.5 (terrain only) | 0.25 | 0 | 0.5 | 0.75 ** | Buildings (+ surfaces) + terrain |
| Buildings + Gr/Gw + trees | False (buildings) + True (trees + buildings facades) | 0.25 | None * | 0.5 (terrain only) | 0.20 | 0.7 | 0.5 | 0.75 ** | Buildings (+ building surfaces) + terrain + trees |
| **Scenario 3: Grass** | | | | | | | | | |
| Buildings only | False | 0.25 | 0 | 1 *** | 0.25 | 0 | 0.5 | 1 | Buildings + terrain |
| Buildings + trees | False (buildings) + True (trees) | 0.25 | None * | 1 *** | 0.20 | 0.7 | 0.5 | 1 | Buildings + terrain + trees |
| Buildings + Gr/Gw | False (buildings) + True (buildings facades) | 0.25 | 0 | 1 (terrain only) | 0.25 | 0 | 0.5 | 1 | Buildings (+ surfaces) + terrain |
| Buildings + Gr/Gw + trees | False (buildings) + True (trees + buildings facades) | 0.25 | None * | 1 (terrain only) | 0.20 | 0.7 | 0.5 | 1 | Buildings (+ building surfaces) + terrain + trees |

* The algorithm evaluates the horizontal area of all "ContextShade" with "True" in "uwg_is_veg" (i.e., trees and green facades in the respective scenarios). ** The value refers only to grass but also considers the 25% of the terrain with the green roofs on buildings. *** Automatically subtracts buildings footprints to calculate the indicated amount of grass cover as a percentage of the remaining terrain.

### 3.5. Experimentation

3.5.1. Scenario 1: Stone Pavement

Considering the reference scenario without green infrastructures, the context modeling considered black stone for pavements and buildings. Therefore, the "uwg_is_veg" was set as "False" and there were no components related to the vegetation. Also, since the Etna volcanic stone is black, regarding the terrain modeling, the "albedo_property" in the DF Terrain component was set as 0.1 (albedo for dark/black surfaces). In addition, the absence of grass and trees determined the "tree_cover" and the "grass_cover" in the UWG workflow component equal to 0.0. Consequently, the vegetation parameters ("veg_par") in the UWG Simulation Parameters component had albedo, tree latent, and grass latent properties equal to 0.0, as well as the vegetation cover percentage ("veg_cover") in the reference .epw file Parameter component. At the beginning of the second phase, which is dedicated to the UTCI calculation, the .epw file from the DF Write .epw component was uploaded, allowing LB to consider the weather conditions within the model built using DF and HB. Finally, regarding the LB HumanToSky Regulation component for the UTCI calculation, the context comprised of the buildings surrounding the human body and the terrain, and also the terrain below the buildings, as the algorithm calculated the difference between the whole model surface and the building footprint.

In the stone pavement and trees scenario, the trees were added. Consequently, the context modeling considered still the buildings only. Differently, another "ContextShade" component was added for the trees and "uwg_is_veg" was set as "True". The outputs were linked to the context in the DF model component. The albedo in the DF Terrain component was left at 0.1. Also, the "tree_cover" had no specific value so that the algorithm could

evaluate the horizontal area of all "ContextShade" geometry with the "True" setting for "uwg_is_veg". Also, in this scenario, the vegetation parameters ("veg_par") albedo was 0.16 (for deciduous plants, typical values are 0.15–0.18), tree latent was 0.7, and grass latent was 0.0. Finally, regarding the LB HumanToSky Regulation component for the UTCI calculation, the context was composed by the surrounding buildings, the terrain, and the trees.

In another scenario with the same stone pavement, the surrounding buildings were covered with green roofs and green walls. Therefore, the context modeling always calculated buildings only, the "ContextShade" "uwg_is_veg" component was set as "True", the pavement albedo in the DF Terrain component was still set as 0.1, and the "tree_cover" property was 0.0 because there are no trees in this scenario. Differently, "wall_albedo" and "roof_albedo" were 0.25 and the roof vegetation percentage was 1.0 in the DF Assign Building UWG properties component to simulate the green walls and roofs. Furthermore, the "veg_par" albedo was 0.25 (vegetation albedo), tree latent was 0.0, and grass latent was 0.5, while the "veg_cover" in the Ref.epwPar component was set as 0.25 (it considers the 25% of the soil occupied by buildings with green roofs). Finally, regarding the LB HumanToSky Regulation component for the UTCI calculation, the context was made of the buildings surrounding the human body and the terrain.

Finally, the scenario containing stone pavement and Gr/Gw and trees saw the integration of all of the properties described in all the previous scenarios, i.e., "ContextShade" "uwg_is_veg" component was "False" for buildings; "uwg_is_veg" was "True" for trees; terrain albedo was 0.1 (stone pavement); ModelUWG "tree_cover" was "None" (it evaluates the horizontal area of all "ContextShade" with "True" in "uwg_is_veg"); grass cover was 0.25 (it considers the green roofs). Also, vegetation albedo was 0.1, tree latent was 0.7, grass latent was 0.5, and vegetation cover was 0.25. Finally, the HumanToSky component considered the "Context" as buildings, terrain, and trees.

### 3.5.2. Scenario 2: Permeable Pavement

The permeable pavement scenario was structured as follows: the pavement color was changed to light (cool material) with grass inside the geometric pattern of the pavement to obtain an albedo equal to 0.25; "ContextShade" "uwg_is_veg" was "False" for buildings (no green wall and roof); terrain albedo (50% white pavement, 50% grass) was 0.25; ModelUWG "tree_cover" was 0.0 (no trees); grass cover was 0.5 because the grass percentage was 50% compared to the whole terrain surface. Also, vegetation albedo was 0.25 (referring to the grass part of the pavement); tree latent was 0.0; grass latent was 0.5; vegetation cover was 0.5 ( to the percentage of grass in the permeable pavement). Finally, the HumanToSky component considered the buildings and terrain as the "Context".

In the permeable pavement and trees scenario, "ContextShade" "uwg_is_veg" was "False" for buildings and "True" for trees; ModelUWG "tree_cover" was "None" (to evaluate the horizontal area of all "ContextShade" with "True" in "uwg_is_veg") and grass cover was 0.5. Also, vegetation albedo was 0.20 (the average of grass and tree albedo (0.25 and 0.16, respectively)); tree latent was 0.7; grass latent was 0.5; vegetation cover was 0.5. Finally, the HumanToSky component considered the "Context" as buildings, terrain, and trees.

In the scenario with permeable pavement and Gr/Gw, the "ContextShade" "uwg_is_veg" was "False" for buildings geometry and "True" for buildings facades (vertical surfaces of the buildings were set as separated geometry); the ModelUWG "tree_cover" was 0.0; grass cover was 0.5 (considering the vegetation on the pavements only), while the vegetation cover was 0.75 (considering the green roofs as 25% of the total surface and the vegetation inside the pavement). Vegetation parameter tree latent was 0.0.

In the last scenario (permeable pavement with Gr/Gw and trees), "ContextShade" "uwg_is_veg" was "False" for buildings geometry, "True" for buildings facades and "True" for trees. ModelUWG "tree_cover" was "None". Finally, albedo was 0.25; tree latent was 0.7, grass latent was 0.5, vegetation cover was 0.75.

### 3.5.3. Scenario 3: Grass

In the grass pavement scenario, the variations in the workflow were the same applied for the scenario with permeable pavement. The factors varied only where the grass was added, as follows: terrain albedo was 0.25 (in this scenario it is grass); ModelUWG "grass_cover" was 1.0; grass latent was 0.5; Ref.epwPar "veg_cover" was 1.0. Also, UWGSimPar "veg_par" albedo was 0.25 (considering the grass as pavement). Regarding the grass and trees scenario, the following parameters were changed: ModelUWG "tree_cover" was "None"; UWGSimPar "veg_par" tree latent was 0.7. UWGSimPar "veg_par" albedo was 0.20 (grass and trees albedo average). The grass and Gr/Gw scenario considered buildings and terrain (the workflow considered the buildings envelope with grass) in the HumanToSky component "Context". Finally, the last scenario (grass and Gr/Gw and trees) was built as an integration of the previous ones, considering grass properties together with Gr/Gw and trees in terms of latent and cover percentage.

## 4. Results and Discussion

Each scenario, i.e., stone pavement, permeable pavement, and, grass comprised three variations (i.e., trees, Gr/Gw, Gr/Gw + trees). Therefore, the results showed 12 outputs in terms of UTCI values to understand the effect of different green strategies on thermal comfort in the built environment.

### 4.1. Scenario 1: Stone Pavement

Figure 5 shows the maximum values of thermal comfort that the human body experiences on the 15 July at 13:00. These are related to the stone pavement scenario by adding the different green infrastructures. The UTCI in the stone pavement scenario with no green infrastructures was equally distributed on the human body, with values of about 36 °C. This is reasonable because the human body is completely exposed to solar radiation and there is no shading or evapotranspiration effect to reduce the temperatures on the body. In addition, the high UTCI is due to the black color (low albedo) of the stone pavement because the infrared radiation of the ground depends on surface temperature and emissivity. In this scenario, the comfort level is minimum, and it is necessary to design green strategies to reduce UTCI.

When trees were added on the square pavement as the first hypothesis to re-design the public space, the results showed a different distribution of the thermal comfort on the human body, with temperatures varying between 31 °C and 32 °C, thus reducing the UTCI by more than 4 °C compared to the scenario with no trees. Because trees are punctual green infrastructures, the part of the human body (in this case, lower parts of the body) that is more shaded by the tree had a lower UTCI, demonstrating a temperature difference of 1 °C. Therefore, it is important to design an efficient tree pattern on the public space to maximize the effect on thermal comfort.

The human body placed on stone pavement but close to a green wall had reduced the body temperatures of about 3 °C (33 °C perceived) compared to the scenario where it is surrounded by dark facades. This outcome is due to the evapotranspiration effect of the green wall. It is important to note that there is no shade provided by the surrounding building because the analysis period was set at 1.00 p.m. as described in the previous section and, therefore, the sunlight directly impacts on the human body. Also in this case, the temperature distribution is not uniform on the human body because the parts of the body that are closer to the green wall have lower temperatures. Consequently, when the human body is far from the green envelope, it has no affect on the thermal comfort.

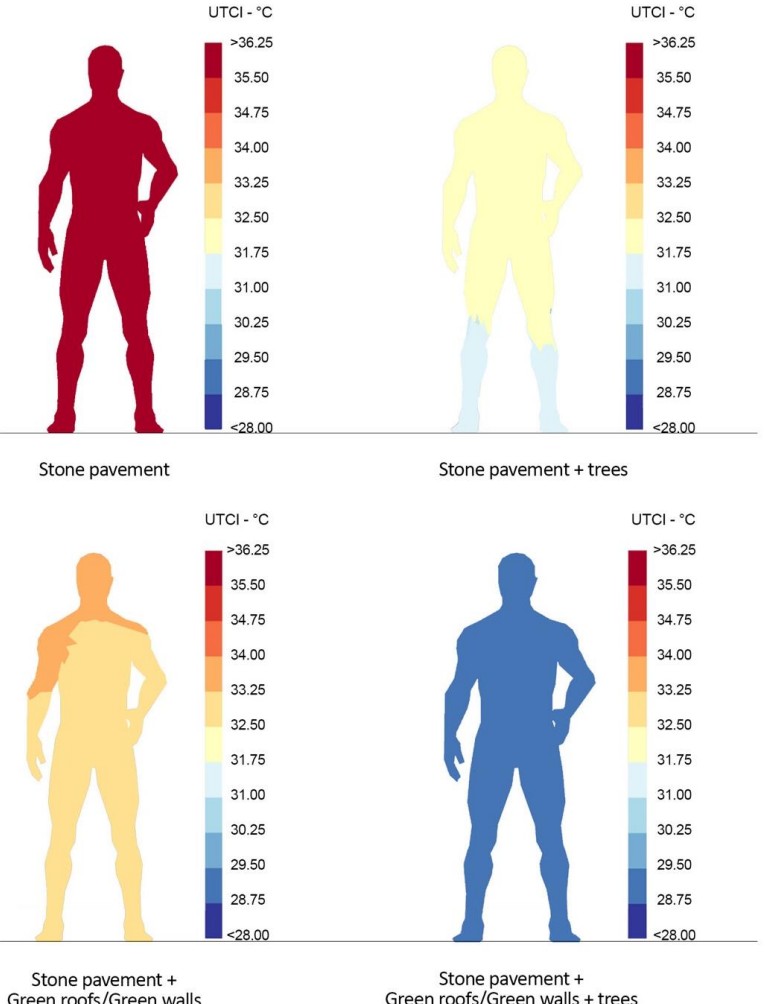

**Figure 5.** Scenario 1: stone pavement.

Finally, in the last scenario with stone pavement, Gr/Gw, and trees, thus integrating all the green strategies, there was a temperature reduction of up to 7 °C under shadow and 6 °C under solar radiation, with the UTCI values varying between 29 °C and 30 °C, demonstrating that an integrated design of multiple green infrastructures can provide high comfort levels in the built environment.

### 4.2. Scenario 2: Permeable Pavement

Figure 6 shows the results of the permeable pavement scenario with the addition of the different green infrastructures. The permeable pavement scenario presented results similar to Scenario 1 in terms of temperature distribution, but differed in terms of UTCI. Because the permeable pavements consist of a pattern comprised of both a cool material and grass, it represents a strategy to reduce the temperatures in the urban context. As described in the methodology section, this pavement type was combined with different green infrastructure strategies to evaluate its contribution to improving outdoor thermal comfort. When only the permeable pavement was installed, the temperatures were uniform on the human body with a UTCI around 34 °C. Compared to the scenario with the stone pavement, the permeable pavement allowed a reduction in temperature of about 2 °C, thanks to the lower albedo of the cool materials and grass and to the evapotranspiration effect of the grass placed inside the pavement.

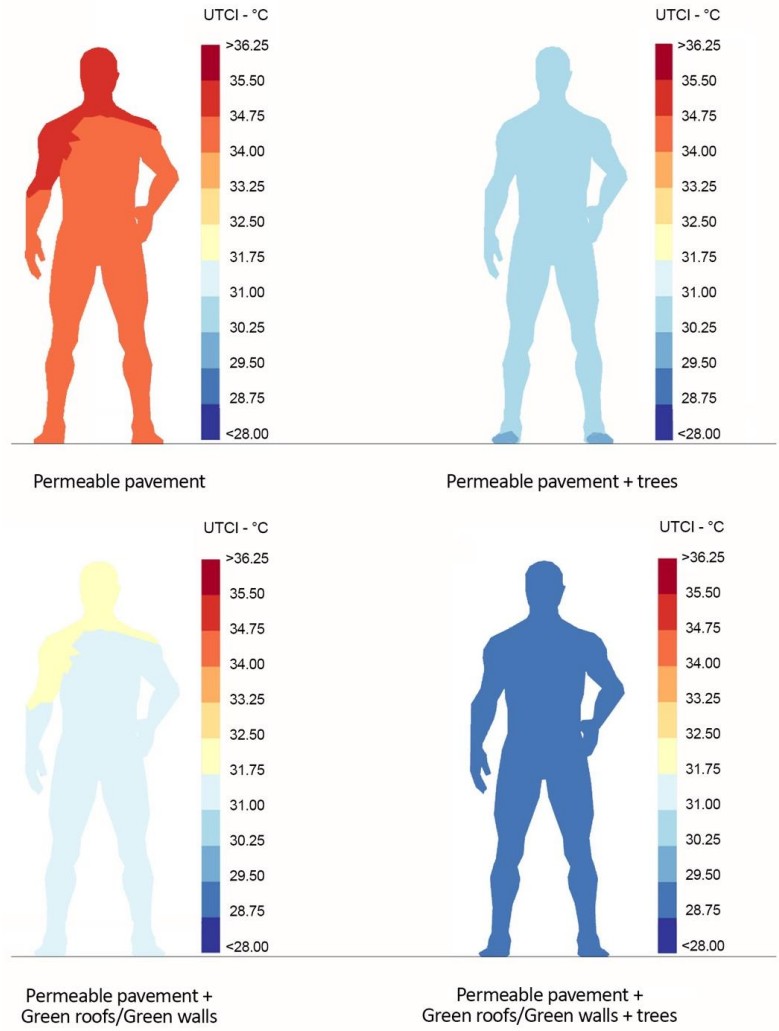

**Figure 6.** Scenario 2: permeable pavement.

Concerning the permeable pavement with trees, the results showed a temperature on the human body of about 30 °C, thus reducing the UTCI by 4 °C compared to the scenario with only the permeable pavement. Also in this case, there is a reduction of 2–3 °C compared to the same situation in Scenario 1. When the human body is placed close to the green wall, the temperature was reduced by another 1 °C, which is up to a 5 °C change from the compared to the same situation in Scenario 1.

Finally, the last situation in the Scenario 2 included the permeable pavement, Gr/Gw, and trees, integrating all the green strategies. In this case, the UTCI was reduced by up to 6 °C, with values around 29 °C. Compared to the results obtained in the corresponding situation in Scenario 1, the UTCI was about 1 °C lower, demonstrating the positive effect of the permeable pavement coupled with the green strategies in providing thermal comfort in the built environment.

### 4.3. Scenario 3: Grass

Figure 7 shows the results of the grass scenario combined with the different green infrastructures. The UTCI with no green infrastructure is uniformly distributed on the human body, and is approximately 34–35 °C. When the trees were added on the grass, the UTCI was 30 °C with slight variations of up to 0.5 °C from shaded to exposed parts of the body. Because grass is more effective in lowering the temperatures on the human body, the effect of adding the tress to the grass is lower compared to Scenario 1 when the trees were added to the stone pavement.

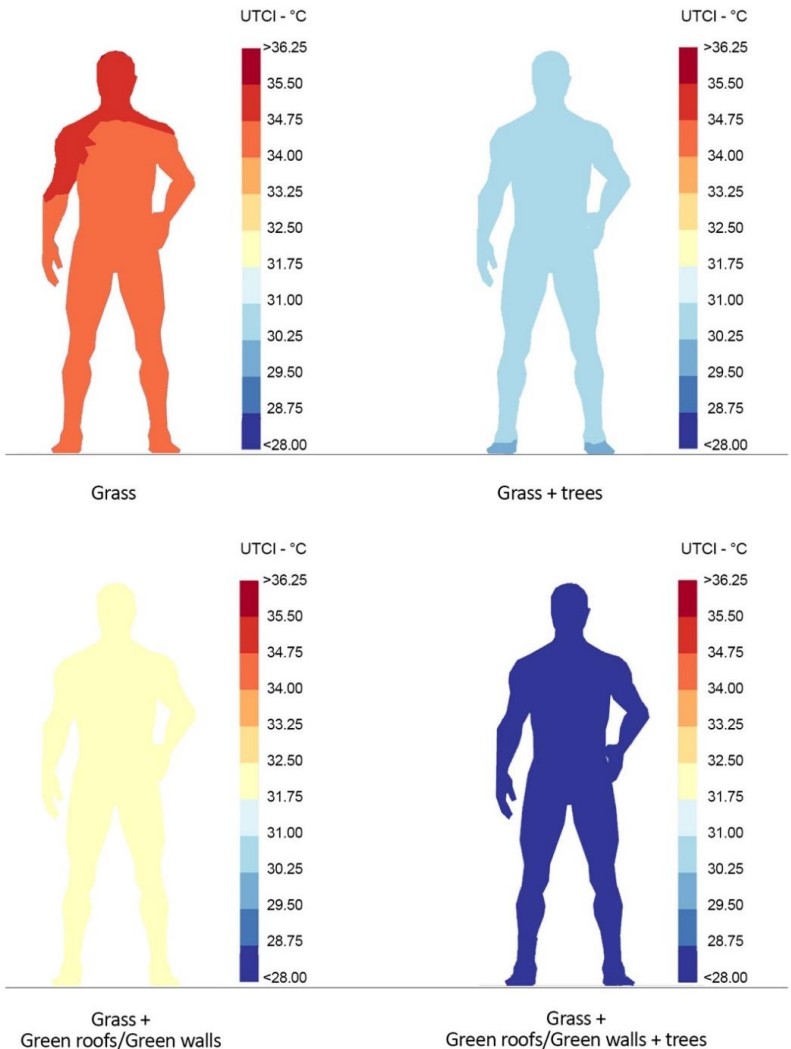

**Figure 7.** Scenario 3: grass.

When the human body was placed close to the green wall in the grass scenario, the UTCI improved by 2–3 °C (around 32 °C perceived) of thermal comfort compared to the situation exposed to the sun and surrounded by dark facades.

Finally, the grass scenario with all the green infrastructure strategies integrated showed a difference in temperature up 6 °C, with UTCI values always around 28 °C. This value was the lowest of all the scenarios, demonstrating the effectiveness of grass integrated with green infrastructures in providing human thermal comfort in the built environment.

The results show that applying nature-based solutions to the urban design can actively alter the thermal comfort of quite large areas (a 2000 m$^2$ square is used in the discussed scenarios), thus improving the environmental quality of life.

Also, major effects are achievable not only by the widespread use of vegetation, but also through an appropriate combination of cool surfaces with vertical or horizontal vegetation. Vegetation in the form of green walls works as a strong element in humidity and temperature regulation, and also for providing shade.

Naturally, direct shadow is the best strategy that can be combined with natural soil. It appears to multiply the effects of thermal comfort, by considerably reducing the feeling of hot temperatures. Consequently, combining punctual and diffuse green infrastructure is one of the best solutions. However, such infrastructure may be difficult to maintain if maintenance is considered as an economic issue. In such situations, the wall vegetation

may be reduced to the base of the building, namely the height near to people, where the benefits can be perceived, even if less intensively.

## 5. Limitations and Future Developments

The model considered different simplifications and schematizations to guarantee reliable values. Some of the limitations are related to the vegetation albedo within scenarios with grass and trees. In fact, in these scenarios, the albedo considered was 0.20, an average of the 0.16 and 0.25 for trees and grass, respectively. Other schematizations are related to the tree shapes and distribution in the built environment. In fact, it was necessary to simplify the geometry as much as possible, because the more realistic the geometry, the more accurate the calculation, the more time that would have been needed for the simulation. Finally, it should be considered that in the green walls and roofs scenarios, the whole vertical and horizontal surfaces were covered in grass, which does not consider windows or other building elements that would reduce the green areas.

In future studies, water can be considered and coupled with different green-blue infrastructures. The contribution of water added to the investigated green infrastructure can alter the perceived temperature and, even more, the thermal comfort.

As economic aspects and maintenance factors are crucial when working with green infrastructures, the research can be implemented with different data by building new Grasshopper components that considering economic factors, depending on defined goals and available resources. Finally, the optimization of the codes can help build awareness with the community, addressing the necessity to learn and take more aware public decisions when enhancing thermal comfort in the built environment.

## 6. Conclusions

In this paper, a digital workflow was developed by integrating existing tools to evaluate how different green infrastructure strategies affect thermal comfort. The workflow was applied to Catania (southern Italy) as typical historical urban context, consisting of a square surrounded by three-floor buildings. The literature review indicated that Grasshopper, Ladybug, Honeybee, and Dragonfly were reliable tools to determine the effect of vegetation on thermal comfort by measuring the UTCI.

Three basic scenarios were created depending on the pavement material used in the built environment: black stone (reference material derived from local volcanic rock), a permeable pavement, or grass. These three scenarios were combined with different green infrastructure strategies: pattern of trees in the square, green walls and roofs on the surrounding buildings, and the integration of all these above-mentioned strategies.

The main results can be summarized as follows:

- The UTCI in the stone pavement scenario with no green infrastructures was equally distributed on the human body, with values of about 36 °C.
- When trees were added on the square pavement, the UTCI was 31 °C, thus reducing the UTCI by more than 4 °C compared to the scenario with no trees, due to the shade provided by the trees.
- A human body placed close to a green wall had a reduced UTCI (33 °C perceived) that was approximately 3 °C cooler than in the first scenario, thanks to the evapotranspiration effect of the green wall.
- The last scenario with a stone pavement, Gr/Gw and trees, thus integrating all the green strategies, saw UTCI values reduced by up to 7 °C and in the range of 29 °C to 30 °C.
- The above-mentioned results were enhanced when permeable materials and, most importantly, when grass were considered as square pavement.
- Combining grass with trees and green roofs and walls can reduce the UTCI by up to 8 °C compared to the reference scenario with black stone materials. In this case, the UTCI was 28 °C.

However, this temperature reduction was highly affected by the location of the human body into the built environment and by the evapotranspiration rate. When the body was far from the tree and the green walls, the UTCI rapidly increases. Similarly, when the vegetation is dry and, therefore, there is no evapotranspiration effect, the UTCI increase as well. Therefore, an accurate design of the built environment is necessary to enhance the positive effect of green infrastructures on thermal comfort.

**Author Contributions:** Conceptualization, S.C.; methodology, A.L.; software, A.L.; investigation, A.L.; data curation, S.C. and A.L.; writing—original draft preparation, S.C. and A.L.; writing—review and editing, S.C. and A.L.; visualization, S.C.; supervision, S.C.; funding acquisition, S.C. All authors have read and agreed to the published version of the manuscript.

**Funding:** The publication was created with the co-financing of the European Union—FSE-REACT.

**Data Availability Statement:** No new data were created or analyzed in this study. Data sharing is not applicable to this article.

**Conflicts of Interest:** The authors declare no conflict of interest.

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
