# Peer review of "Thermal Comfort in the Built Environment: A Digital Workflow for the Comparison of Different Green Infrastructure Strategies"

_atmosphere, doi:10.3390/atmos14040685_

Round 1

Reviewer 1 Report

The current research has developed a digital work flow integrating existing tools to evaluate the influence of green infrastructure to the thermal comfort. This is an interesting study that providing a useful method to give scientific evidence when make  greenery design. The research results have been  discussed, while the author failed to give specific strategies, which is considered important for the local government and designers.

Suggestions:

1. The relevant references is insufficient. There are only seven references on green infrastructure and human thermal comfort that are cited, which would lead to the lack of solid background in introduction part.

The author needs to make a more specific clarification what is green infrastructure? Which is also important in the scale of urban planning and urban design, such as: 10.1016/j.scitotenv.2022.155307

2.Page 2, Line 69. Does UTCI needs clothing level and metabolic rate for its calculation?

3. A methodology diagram should be given.

4 The diagrams presented are not clear enough, nor do they illustrate the module functions that the authors used, such as Figure 3- Figure 11.

5. How do you calculate outdoor UTCI? More details should be described. Was wind speed included in the calculations?

6. In figure 13is it an average distribution or a maximum value ?

7. The design strategies should be given according to the research results.

Reviewer 2 Report

 1.     The main purpose of this study on the digitalization for improving sustainability and the human comfort in the built environment would be definitely a significant approach to thermal comfort in urban areas in the near future.  

2.     In terms of methodology, it is reasonable to consider variable conditions, including the square pavement type, the tree pattern, the green roof and wall of the building envelope and the integration of all these strategies.

3.     Yet the Parameters in these three scenarios for the components used in the workflow seem lack physical factors, which represent the causes for the thermal discomfort, so that one can develop appropriate improvement measures.

It is suggested that the authors explain and supplement how physical parameters are considered in the workflow, such as solar radiation, thermal capacity of the pavement (as well as the wall), air humidity, soil moisture content and its evaporation rate etc., which are essential for the accurate calculation in a digital workflow and visual programming. 

4.     The solar radiation includes at least visible light and infrared, which transmit heat in major, please explain how infrared affects UTCI with the black color (low albedo?) of the stone pavement.

5.     In addition, an important factor for the thermal environments would be the wind fields. Please explain how the wind fields or wind speed are considered in the workflow.

Round 2

Reviewer 1 Report

The authors had made corresponding revise according to the commet I made. I have no other commet.